# Seasonal Changes in the Antioxidant Activity and Biochemical Parameters of Goat Milk

**DOI:** 10.3390/ani13101706

**Published:** 2023-05-21

**Authors:** Oksana A. Voronina, Sergei Yu. Zaitsev, Anastasia A. Savina, Roman A. Rykov, Nikita S. Kolesnik

**Affiliations:** Federal Research Center for Animal Husbandry Named after Academy Member L.K. Ernst, Dubrovitsy 60, Podolsk Municipal District, Moscow 142132, Russia; kirablackfire@mail.ru (A.A.S.); brukw@bk.ru (R.A.R.); kominisiko@mail.ru (N.S.K.)

**Keywords:** biochemistry of goat milk, antioxidants and vitamins of goat milk

## Abstract

**Simple Summary:**

Goat milk is an excellent product, which contains a large amount of nutrients and biologically active compounds. The action of many of them also manifested in antioxidant activity. However, the composition of goat’s milk varies from season to season, which may affect its functional activity. To study this effect was the purpose of our work. The numerous significant changes in the biochemical parameters of goat milk and its antioxidant activity depending on the season of the year found. This main result of our work will be useful for goat physiology and biochemistry, as well as for the monitoring of the quality of goat’s milk.

**Abstract:**

Goats are ubiquitous, including in hot and dry regions, while also being very sensitive to climate fluctuations, expressed in temperature differences. This affects their productivity and milk quality. Adaptation to heat requires high energy costs, affects “neurohumoral” regulation and is accompanied by oxidative stress with the increased production of free radicals. The aim was to study the main biochemical parameters of goat milk and its antioxidant activity depending on the season of the year. Sampling was carried out in April, June, August and October. Analysis of the biochemical components and antioxidant activity of goat milk was performed using modern analytical systems. From spring to autumn, the mass fraction of true or crude proteins in goat milk increased by 14.6–63.7% or by 12.3–52.1%, and the mass fraction of caseins also increased by 13.6–60.6%. For vitamin C level and the total amount of water-soluble antioxidants, a pronounced gradual decrease from spring to autumn was observed. In the summer period, a small increase in the carotene level in milk (by 3.0–6.1% compared to April) was established. Vitamin A content increased by 86.5% (June) or by 70.3% (October) compared to April. Thus, the numerous significant changes in the major parameters of goat’s milk depending on the season were revealed.

## 1. Introduction

Goat milk is an excellent product that contains a large amount of nutrients and biologically active compounds [1]. The action of many of them is also manifested in antioxidant activity [2]. The greatest contribution to the antioxidant activity of goat milk is made by polyphenolic compounds, various amino acids, vitamins A, E and D, carotenoids, calcium, zinc, selenium, some enzymes (superoxide dismutase, catalase, glutathione peroxidase), oligosaccharides and bioactive peptides [1,2]. Goat’s milk contains less lactose and more protein, fat, calcium and vitamin A compared to cow’s milk [2,3]. Despite the ubiquitous distribution of goats, including in hot and arid regions, they all have to adapt to heat stress in spite of some breed differences [4,5,6]. Although goats and sheep are better able to adapt than cattle, they are more susceptible to the effects of climate fluctuations and temperature changes [7].

A number of adaptive physiological processes are activated during exposure to high temperature. This is expressed in changes in behavioral reactions, such as a decrease in the consumption of dry matter and an increase in water consumption. This leads to a change in microbial profile and a decrease in rumen activity, followed by a loss of productivity [6,7,8]. During the changes in humoral regulation (along the hypothalamic–pituitary–adrenal axis), hormones (prolactin, cortisol, triiodothyronine, thyroxine) perform thermoregulatory adjustments, directing metabolic reactions adjusted for heat stress [7,8]. Thyroid hormones play a special role in this process since a decrease in the synthesis of triiodothyronine and thyroxine at elevated temperatures leads to an increase in the intensity of oxidative processes in cells. This leads to an increase in oxygen consumption and cellular heat production, which increases the basal metabolic rate [6,8]. The redistribution of blood flow in this case is directed to a greater extent to the periphery, due to which there is a loss of excess heat. Adaptive rearrangements require high energy costs, which leads to a decrease in productivity and a decrease in the content of fat, protein, lactose and some other indicators of milk quality [8,9]. Thus, the dominant of “productivity” is replaced by “the dominant of adaptation to elevated temperature” [7,10].

Heat stress (as a non-specific physiological response to any needs of the body at high ambient temperatures [10]) is strongly associated with oxidative stress and the increased production of free radicals [7]. The influence of temperature, relative air humidity and the duration of solar radiation affects the “lactation curve” and indicators of the quality of goat milk, such as the mass fraction of fat, mass fraction of protein, lactose, etc. [4,9,10]. For example, the correlation coefficients between air temperature and quality indicators are the following: “−0.90” (*p* ˂ 0.05) for the mass fraction of fat, “−0.77” (*p* ˂ 0.05) for lactose and, “−0.74” (*p* ˂ 0.05) for the mass fraction of protein [4].

There is low activity in terms of glutathione peroxidase, superoxide dismutase [10] and acidic and alkaline phosphatase [8], as well as low concentrations of vitamins C and E, coinciding with the high production of peroxide and hydroxyl radicals in the blood serum of goats during heat stress [10]. In summer, the contents of vitamins A and E are higher in milk; the proportion of saturated fatty acids is also higher compared to unsaturated fatty acids. This has a direct influence on the properties of milk and cheese made from it [11].

Some antihypertensive, anti-inflammatory, antitumor, antioxidant, antibacterial, immunomodulating and immunostimulating properties of goat milk have been shown in previous studies [1,12,13,14]. To our regret, these milk properties in those studies [1,12,13,14] are considered separately from the conditions of production and the influence of the season regarding the quality of the resulting product. The authors of [15,16] emphasize the importance of the following factors: the breed and lactation stage of goats, as well as their influence on the quality of goat milk. In [4,5,6], the authors study and emphasize the propensity of goats to heat stress, and the authors of [17,18] show the universal effect of heat stress on goats, regardless of their breed. The influence of the season on the nutritional properties and technological qualities of sheep and goat milk has also been shown in previous studies [19,20,21]. Additionally, the work of Chávez-Servín J.L. [22] considers the influence of feeding systems, heat treatment and seasonal effects on phenolic compounds and the antioxidant capacity of goat milk, whey and cheese. Thus, there is potential importance for the seasonal characteristics of goat milk properties.

The optimal temperature zone for goat farming, according to Hein L. et al. [23], is about 12–24 °C, and heat stress occurs when the body’s heat production and release balance is disturbed, which affects animal performance [23,24]. According to Collier R.J. [25], activation of adaptive systems is initiated at skin surface temperatures exceeding 35 °C. This leads to the activation of a highly conserved protein cascade and changes in gene expression.

In this regard, we have considered that it is important to study the total activity of water-soluble antioxidants (TAWSA) in goat milk, the activity of vitamins (A and C) and carotene, as well as the biochemical composition of goat milk, which is obtained in some seasons of the year with a temperature difference of about 15 °C. The novelty of this research is associated with an integrated approach to the study of the biochemical and antioxidant composition of goat milk depending on the season for various breeds of goats, including local breeds. The advantage of this work lies in its value for climatic regions with similar temperature cycles or temperature differences. The relevance of this work is associated with changes in climatic conditions and increased interest in the adaptation of productive animals to these changes.

The purpose of this work was to study the antioxidant activity of goat milk in relation to changes in the main biochemical parameters depending on the season.

## 2. Materials and Methods

### 2.1. Climatic Conditions

The Moscow region is a region with a temperate continental climate, which is characterized by moderately frosty winters and warm summers [26]. According to the site http://www.pogodaiklimat.ru/ (accessed on 12 May 2023), the average monthly temperatures for the Voskresensky district of the Moscow region in 2022 were the following: January −6.1 °C, February −2.0 °C, March −1.7 °C, April +6.0 °C, May +10.5 °C, June +18.5, July +20.6 °C, August +21.6 °C, September +9.9 °C, October +7.0 °C, November −0.8 C, December −4.7 °C. Goat milk sampling was carried out in April, June, August and October. During the entire observation period, the animals had free access to walking, and from May to September they grazed with access to shaded areas. The height of the Voskresensky district of the Moscow region is about 122 m above sea level.

### 2.2. Goat Milk Sampling

Goat milk sampling was carried out during spring and autumn (April and October) as so-called “reference” seasons, in comparison to the summer months (June till August) in the Moscow region in 2022. The samples were taken in accordance with the rules for sampling milk (GOST 26809.1–2014.). The age of the animals was 2–3 years, and the duration of lactation for the period of sampling was 3–5 months. The sampling periods were April (n = 12), June (n = 12), August (n = 24) and October (n = 27). Milk samples were obtained each time from the following goat breeds: Alpine, Nubian, Czech and Russian. There were 3 representatives of each breed in the groups of April and June, 6 in August, and 7 goats of the Alpine, Nubian and Czech breeds and 6 goats of the Russian breed in October. Animals were selected for the study while taking into account the state of their health. Before combining animals of different breeds into groups for statistical processing by season according to similar age and lactation period, we excluded the significance of differences in breeds, age and month of lactation. After general statistical treatment, a comparison was made across the seasons. Depending on the timing of obtaining samples, we considered and compared 4 groups: April, June, August and October. Statistical processing of the obtained results was carried out in “Microsoft Excel” using the “Data Analysis” package and “R” using the “Psych” package. The significance of differences between breeds, age and lactation terms was assessed by the Mann–Whitney U test, with the significance of differences between seasons assessed using the critical values of a Student’s *t*-test.

### 2.3. Measurement of the Total Amount of Water-Soluble Antioxidants

The amperometric method was used to study the total amount of water-soluble antioxidants (TAWSA) [27]. Measurements were made on the “Tsvet-Yauza 01-AA” instrument (Russia). The TAWSA parameter was determined by measuring the strength of the electric current that occurs during the oxidation of molecules on the electrode surface at a certain potential. The TAWSA parameter was measured in terms of equivalence to gallic acid. For this, “working solutions” were prepared for calibration from a solution of gallic acid (100 mg/dm^3^) with a mass concentration of 0.2, 0.5, 1.0 and 4.0 mg/dm^3^. Orthophosphoric acid solution at a concentration of 2.2 mM (mmol/dm^3^) was used as an eluent.

### 2.4. Determination of Vitamin A and Carotene according to the Method of Kondrakhin I.P. [28]

Vitamin A and carotene were determined by a spectrophotometric method based on the ability of petroleum ether to extract vitamin A and carotene [28]. A volume of 2 mL of milk was poured into a test tube and 0.2 mL of KOH was added, with the mixture then left for saponification for 2 h. Next, hydrolysis was carried out in a water bath for 40 min. After that, 2 mL of alcohol was added to the cooled flask and transferred to a separating funnel, where it was shaken and then extracted with 10 mL of petroleum ether. The ethereal extracts were washed until the traces of alkali disappeared. The washed extract was transferred to a flask where it was dehydrated with sodium sulfate. The measurements were performed on a PE-5300V spectrophotometer (Ekros Research and Production Association) at wavelengths of 325 nm for vitamin A and 452 nm for carotene.

### 2.5. Determination of Vitamin C in Milk with 2,6-Dichlorophenolindophenol

A volume of 25 mL of milk was measured into a flask, and proteins were then precipitated through addition of 2 mL of a saturated solution of oxalic acid and 5 mL of a saturated solution of sodium chloride. The protein precipitate was filtered off through a dry paper filter. The filtrate was collected in a clean flask. A volume of 5 mL of the filtrate containing vitamin C was measured into three flasks, and the filtrates in all flasks were titrated with a 0.001 N solution of 2,6-dichlorophenolindophenol. The end of filtration was determined by the appearance of a faint pink color, which persisted for 1–2 min. Calculation of vitamin C concentration (mg per 100 g of milk) was carried out according to the average value from 3 sampling flasks.

To calculate the content of vitamin C ([X]), the following formula was used: [X] (mg/100 g) = 0.088 * A * B * 100/D * E
where 0.088 is the amount of ascorbic acid (mg) corresponding to 1 mL of a 0.001 N solution of 2,6-dichlorophenolindophenol, A is the volume of the solution of 2,6-dichlorophenolindophenol used for titration (mL), B is the total extract volume (mL), D is the sample weight (g), and E is the recalculation per 100 g of the product [29].

### 2.6. Determination of the Biochemical Parameters of Milk

Analysis of the biochemical parameters of milk was carried out at the Center for the Collective Use of Scientific Equipment “Bioresources and Bioengineering of Farm Animals” L.K. Ernst. The analysis was performed using the MilkoScan 7/Fossomatic 7 DC analytical system (Denmark) and MilkoScan 7, a spectrophotometer based on Fourier transform infrared spectrophotometry [30].

## 3. Results

### 3.1. Biochemical Composition of Goat Milk Depending on the Season

The numerous biochemical and other parameters of goat milk depending on the season were measured (Table 1).

When analyzing the biochemical parameters of milk, numerous variations in the average values for these four groups were established (Table 1). The changes in the mass fraction of fat compared to April were found as the following: −8.2% (June), −4.8% (August), 12.3% (October) (Table 1). Essential increases in the mass fraction of true protein (MFTP-1) were observed compared to April: 14.6% (June), 18.7% (August) and 63.7% (October). Almost the same increase in the mass fraction of crude protein (MFCP-2) was also found, namely increases of 12.3% (June), 15.4% (August) and 52.1% (October) compared to April. The same tendency was found for caseins, with increases of 13.6%, 15.8% and 60.6% from spring to autumn (Table 1).

A pronounced increase in dry skimmed milk matter (DSMM) was found compared to April, with increases of 3.1% (June), 2.7% (August) 19.3% (October). An increase in total dry matter (TDM) was also observed, with increases of 0.7% (June), 0.1% (August) and 15.5% (October) compared to April. A very small decrease in the lactose content of milk was found, with decreases of −4.1% (June), −6.0% (August) and −4.8% (October) compared to April.

A “huge” decrease in the acetone content of milk was found, namely−66.7% (June), −46.7% (August) and almost complete acetone absence (October) compared to April. The same decrease in the betahydroxybutyrate (BHB) content of milk was found, with −83.3% (June), −50.0% (August) and almost complete BHB absence (October) compared to April.

An interesting change in the urea content of milk was observed, with changes of 5.3% (June), −11.3% (August) and 16.3% (October) compared to April. A small but continuous decrease in the freezing point of milk was exhibited, with changes of 1.3% (June), 2.7% (August) and 3.7% (October) compared to April. There were also very small changes in the pH of milk, namely −0.5% (June), −1.1% (August) and 0.2% (October) compared to April.

### 3.2. Antioxidant Composition of Goat Milk Depending on the Season

The total antioxidant activity of goat milk and the activity of some antioxidants changes with the season (Figure 1a–d).

From spring to autumn, there was a gradual decrease in vitamin C concentration in goat milk, specifically 2.8% in June, 8.1% in August and 12.8% in October (Figure 1a). In addition, there was a decrease in TAWSA activity of 10.6% in June, 15.1% in August and 28.2% in October (Figure 1b). At the same time, there were no reliable differences between the groups, both for vitamin C and for TAWSA. In contrast, significant differences were found for vitamin A content (Figure 1c). In the summer months, there was an increase in its content in comparison with spring data, with an increase of 86.5% in June, 75.7% in August and 70% in October compared to April. The level of carotene in the summer period increased slightly by 6.1% in June and 3.0% in August but decreased by 7.6% in October compared to April (Figure 1d). There were no reliable differences between groups in relation to carotene.

### 3.3. Correlations between Biochemical Parameters and the Antioxidant Composition of Goat Milk

Since the processes of interaction between lipids, proteins, carbohydrates, the low molecular weight compounds of milk and the composition of the antioxidants we studied and the total antioxidant activity of milk are characterized by high complexity, we assumed that their nature largely depends on the amount of each component and can therefore vary with seasonal changes. In order to better understand and describe the possibilities of the mutual influence of the components on each other, we performed correlation analysis. The most interesting results for antioxidants, in our opinion, are described below (Figure 2).

The following important correlations were established in April (Figure 2a): (a) for TAWSA with carotene (0.17), vitamin A (0.32) and C (0.45), urea (0.43) and protein (0.15); (b) for carotene with vitamin A (0.70), vitamin C (0.12), urea (0.33), casein (0.20) and protein 0.26; (c) for vitamin A with vitamin C (0.33), urea (0.54), casein (0.44) and protein (0.43); and (d) for vitamin C with lactose (−0.36). The following reliability was confirmed: true, protein and urea, 0.55 (r^2^ = 0.30, *p* ˂ 0.01); true, protein and vitamin A, 0.43 (r^2^ = 0.18, *p* ˂ 0.05); caseins and urea, 0.48 (r^2^ = 0.42, *p* ˂ 0.05); caseins and vitamin A, 0.44 (r^2^ = 0.19, *p* ˂ 0.05); urea and vitamin A, 0.54 (r^2^ = 0.29, *p* ˂ 0.05); urea and TAWSA, 0.43 (r^2^ = 0.18, *p* ˂ 0.05); vitamin C and TAWSA, 0.45 (r^2^ = 0.20, *p* ˂ 0.05); vitamin A and carotene, 0.70 (r^2^ = 0.49, *p* ˂ 0.05).

The following important correlations were established in June (Figure 2b): (a) for TAWSA with carotene (−0.35), vitamin C (0.31), urea (0.29) and lactose (−0.23); (b) for carotene with vitamin A (0.88), casein (0.18) and protein (0.24); (c) for vitamin A with vitamin C (0.26), urea (0.24), lactose (−0.43) and protein (0.26); and (d) for vitamin C with urea (0.30), casein (−0.36) and protein (−0.37). The following reliability was confirmed: true, protein and urea, 0.48 (r^2^ = 0.23, *p* ˂ 0.05); lactose and vitamin A, 0.43 (r^2^ = 0.18, *p* ˂ 0.05); caseins and urea, 0.43 (r^2^ = 0.18, *p* ˂ 0.05); vitamin A and carotene, 0.88 (r^2^ = 0.77, *p* ˂ 0.001).

The following important correlations were established in August (Figure 2c): (a) for TAWSA with carotene (0.23), vitamin A (0.18), vitamin C (−0.30), casein (0.41), lactose (0.2) and protein (0.41); (b) for carotene with vitamin C (0.16), urea (−0.19), casein (0.53) and protein (0.55); (c) for vitamin A with urea (0.30) and lactose (−0.29); and (d) for vitamin C with lactose (0.24). The following reliability was confirmed: true, protein and carotene, 0.55 (r^2^ = 0.30, *p* ˂ 0.01); caseins and carotene, 0.53 (r^2^ = 0.28, *p* ˂ 0.01); caseins and SCVA, 0.41 (r^2^ = 0.17, *p* ˂ 0.05).

The following important correlations were established in October (Figure 2d): (a) for TAWSA with vitamin C (0.28) and urea (0.26); (b) for carotene with vitamin A (0.87), vitamin C (0.17) and urea (−0.17); (c) for vitamin A with vitamin C (0.17), casein (0.19) and lactose (−0.25); and (d) for vitamin C with casein (−0.31), lactose (0.44) and protein (−0.32). The following reliability was confirmed: true, protein and lactose, −0.64 (r^2^ = 0.41, *p* ˂ 0.001); casein and lactose, 0.62 (r^2^ = 0.38, *p* ˂ 0.001); lactose and vitamin C, 0.44 (r^2^ = 0.19, *p* ˂ 0.05); carotene and vitamin A, 0.87 (r^2^ = 0.76, *p* ˂ 0.001).

## 4. Discussion

### 4.1. Comparability of Our Results with the Data Obtained Earlier

As the main results of this work, the numerous and significant changes in the parameters of goat milk depending on the season were revealed. For example, the mass fraction of proteins (both MFTP-1 and MFCP-2) in goat milk increased by 12.3–18.7% in summer compared to spring. A similar trend (high values for the mass fraction of goat milk proteins in summer) was described in the works of Kljajevic N.V. et al. [4], Li S. et al. [19] and Margatho G. et al. [20]. However, the huge increase in the MFTP-1 and MFCP-2 parameters in goat milk in autumn (in October by 52.1 and by 63.7% compared to April and by 37.9 and by 31.8% compared to August) was found here for the first time. These data (an increase in goat milk protein content in autumn) correlated with heat stress at the particular goat lactation periods. In addition, it is possible to highlight now that this direct tendency is reliable and reasonable. For example, the same tendency was found for casein content in goat milk in autumn, with an increase in October of 60.6% compared to April and 38.7% compared to August. This is especially important because caseins are the major milk proteins by number (their amount is about 80% of total milk proteins) and importance (both structural and functional) [4].

It is important to highlight that the same trends were found in other studies [19,20,21]. For example, in [20], the values for protein mass fractions were at the level of about 3.5% in June and July [20]. For DSMM throughout the year, we obtained values from 8.17% to 9.75%, which are somewhat lower than the values of dry skimmed milk matter (obtained in [21]) where the following values were established: 10.34% for autumn, 10.28% for winter and 9.98% for spring [21]. In spite of some quantitative differences, the seasonal minimum of the value of this indicator in the spring period was established both in our work and in the one described above [21]. In [23], the values of dry skimmed milk matter changed from 8.47% to 9.47%. These data are comparable with our results (Table 1).

The fat mass fraction of goat milk from spring to autumn decreased by 5–8% in summer months and increased by 12% in the autumn period (Table 1), but without reliable “certainty”. It is obvious that goats are more mobile in summer when grazing than in other seasons. Therefore, a slight decrease in milk fat is fully justified in summer, as well as its increase in autumn. According to data from the literature for the milk of goats from New Zealand, the lowest mass fraction of fat (3.79 ± 0.39%) was found in the summer period compared to other seasons [19]. The maximum values for the fat mass fraction were established in January (6.24%) and October (6.75%) [20]. Our data, mentioned above (Table 1), confirmed the previously established general patterns described by other authors [19,20].

The maximal changes in lactose were in the range of 4.1–6.0% (i.e., a small decrease from April to October). This is normal for a seasonal variation and in full agreement with the literature data [19,20,21]. Acetone and betahydroxybuterate (BHB) content in milk was present in trace amounts (Table 1). As to why a significant decrease in acetone and BHB content in summer and autumn is negligible, first of all, the almost complete absence of acetone and BHB content in milk in October, as compared to April, is a good sign for goat physiology. In addition, acetate and BHB in the mammary gland serve as substrates for the synthesis of short-chain fatty acids (C4…C10) “de novo” [23]. They serve, together with urea, as markers of negative energy balance and ketosis [30]. According to the data from the review by Kumar H. [31], their content in goat milk ranges from 0.09 to 0.26 g/100 g.

When analyzing the effect of the change in seasons on the biochemical composition of milk, the authors of [4] obtained significant correlations for fat and protein mass fractions, lactose and some other indicators with air temperature, relative humidity and the duration of solar radiation. For example, for air temperature and MFF, the correlation coefficient was −0.90 (*p* ˂ 0.05), for lactose → −0.77 (*p* ˂ 0.05), and for proteins → −0.74 (*p* ˂ 0.05) [4]. Kljajevic N.V. et al. [4] attribute the obtained results to the low fat synthesis and high water consumption of goats in summer, as well as the high sensitivity of the species to heat stress. At the same time, such indicators as lactose, freezing point and pH essentially did not change from season to season for all samples (n = 75), and the coefficients of variation for these parameters were 7.36%, 2.49% and 2.15%, respectively. Other authors found almost the same features and paid special attention to these trends and parameters [4,19,20]. In our opinion, the last three indicators turned out to be the most stable relative to seasonal fluctuations.

As for the antioxidants in goat milk, some authors point out that ascorbic acid has the greatest potential among the all antioxidants in milk [1,32]. These and other authors note the antioxidant activity of milk protein fractions, especially their hydrolysates [33], or underline the important role of phenolic compounds [22]. We studied the concentration of vitamin C and determined the total amount of water-soluble antioxidants (TAWSA) in milk in terms of equivalence to referential gallic acid [28]. The correlation coefficients between TAWSA and vitamin C were the following: in April, –0.45 (r^2^ = 0.20, *p* ˂ 0.05); in June, 0.31 (r^2^ = 0.10); in August, −0.30 (r^2^ = 0.10); in October, 0.28 (r^2^ = 0.08); however, there were no reliable differences in these indicators compared to the season. As underlined above, proteins, in particular caseins, are noted as a “powerful” antioxidant component of milk [1]. In our study, we obtained a positive and relatively strong correlation (0.41 (r^2^ = 0.17, *p* ˂ 0.05)) between protein content and TAWSA in August.

In addition, we would like to underline the following: a relatively slight decrease was observed in the concentration of vitamin C and TAWSA (2.8% and 10.6% for June, 8.0% and 12.8% for August and 15.1% and 28.2% for October), as well as a significant increase in MFTP-1, including caseins (by 14.6%, 18.7% and 63.7% compared to April). The obtained data are in agreement with the changes in the physiological parameters of goats but require further analysis.

The optimum temperature zone for goats is around 12–24 °C. Heat stress occurs when the balance of body heat production and its return is disturbed, which affects productivity [27]. According to the Hydrometeorological Center of Russia, the average temperatures in April and October in 2022 were at the level of 12 degrees. Therefore, we use these months as a control in relation to the summer months, when the average temperature rises to 25 or more degrees.

The authors, who conducted experiments with goat milk that underwent heat treatment and which was obtained in seasons with different amounts of weather precipitation, noted that the concentration of phenolic compounds in goat milk was higher in unpasteurized samples from the dry season compared to pasteurized samples from the rainy season [22]. In the same work, free-range grazing was found to be a good option for producing higher concentrations of phenolic compounds and higher antioxidant capacity in goat milk [22].

### 4.2. Physiological and Biochemical Mechanisms of the Adaptation of Goats to Heat Stress

The evolutionary mechanisms of goat adaptation to high ambient temperatures include several stages at different levels of matter organization. The mechanisms of physiological and humoral regulation have been described in detail in the introduction to the publication. For a more detailed study of them, one can refer to [6,7,8,34,35].

At the cellular level, heat stress is closely associated with oxidative stress [7], which causes damage to biological molecules. On the one hand, this induces oxidative damage to cells and activates apoptotic pathways. On the other hand, it helps cells survive by increasing the expression of heat resistance genes, followed by an increase in the level of heat shock proteins, antioxidant activity, etc. [36]. The further life of a cell directly depends on the rate of its adaptation to changes induced by heat shock. A number of studies have shown that heat shock proteins such as HSP70 are an ideal biological marker of heat stress in farm animals, including goats [25,36].

The triggering of cascade reactions starts when the skin surface temperature exceeds 35 degrees Celsius [25]. A highly conserved cascade of protein activation and altered gene expression includes activation of heat shock transcription factor 1, increased expression of heat shock proteins (HSPs) and decreased expression and synthesis of other proteins. This may be one of the reasons for the decrease in the level of proteins in milk obtained in the summer. Higher expression of HSP70 mRNA in tissues is a sign of higher “thermotolerance” [36]. In our case, there is a significant decrease in the mass fraction of proteins, in particular goat milk caseins in the summer. Apparently, one of the possible explanations for this is an increase in the synthesis of heat shock proteins. This assumption requires careful verification, in particular the analysis of heat shock proteins, both in milk and in blood serum.

At the same time, increased glucose oxidation [25,37], which accompanies adaptation to heat stress, does not affect the level of lactose in milk. Its level is very stable regardless of temperature. However, the lowest value is observed in August (4.35 ± 0.07%), the hottest month from those presented in this paper. In our opinion, this is due to the special important role of lactose in milk. After all, it is lactose that regulates the level of milk production (by controlling the flow of water into milk) and serves as an indicator of the oxidative status of the mammary gland [38,39]. In addition, lactose serves as a building block for a number of oligosaccharides that appear to play a critical role in regulating growth and development across the gastrointestinal microbiome [39]. The key determinants of lactose synthesis in the mammary gland are α-lactalbumin and β-1,4-galactosyltransferase-1, and their expression in the mammary gland epithelium is tightly regulated by critical hormones, including prolactin, glucocorticoid, insulin, triiodothyronine and epidermal growth factor [39]. The results obtained allow us to conclude that this regulatory pathway of lactose synthesis has a rigid fixation at the genetic level [39]; Maintaining a stable level of lactose in milk during heat stress is no less significant than turning on the synthesis of heat shock proteins.

The regulation of milk fat synthesis also occurs through molecular regulatory networks [40]. The process includes the synthesis of fatty acids and triglycerides, the formation of fat droplets and the absorption and transport of fatty acids. Goat’s milk fat contains more short-chain, medium-chain and unsaturated fatty acids than cow’s milk. This indicates a unique network in fat synthesis from goat milk [40]. Studies [40] show that miR-23a, miR-27b and miR-103 selectively regulate mRNA expression associated with milk fat synthesis. A slight decrease in the level of milk fat in the summer period, which we noted, is not confirmed by statistical significance. Perhaps the regulation of the milk heat synthesis pathway is less affected than that of milk protein synthesis. Testing this assumption requires a lot of work that we plan to do in the near future. Coming back to the level of animal physiology, we once again emphasize such adaptive behavioral reactions as a decrease in appetite, an increase in water consumption, frequent shallow breathing and a shift in gas homeostasis. This is also a large area for further research.

Temperature, as an environmental factor, acts everywhere, affecting entire ecosystems. Therefore, when studying goat milk obtained during the dry period [22], researchers find, among other things, traces of adaptation to heat as part of the plant survival strategy in the form of an increase in the concentration of phenolic compounds in milk. These are non-specific compounds, the synthesis of which does not occur in animals; however, this does not prevent their use in the neutralization of free radicals. In our work, investigating antioxidant activity, we associate the observed changes with seasonal feeding since, during the period from May to September, the animals had access to fresh vegetation. This is clearly seen in the significant changes in the content of vitamin A and, albeit insignificant, in the increase in the level of carotene in the summer. By continuing experiments and observations in this direction (mainly TAWSA and vitamin parameters) and making the knowledge gained available to both producers and consumers of milk and dairy products, the potential benefits of a particular product can be emphasized.

## 5. Conclusions

Thus, the numerous significant changes in the parameters of goat milk depending on the season were revealed. For most parameters, the observed changes are consistent with the general seasonal changes in goat physiology and biochemistry. The relationship between the antioxidant capacity of animal milk and physiology is especially valuable for monitoring of the quality of goat milk.

Thus, our study demonstrated a change in the biochemical and antioxidant composition of goat milk under the influence of natural seasonal temperature fluctuations in the conditions of the Moscow region. Moreover, the valuable indicator TAWSA was studied for the first time in relation to goat milk. Among the studied constituent components of milk (proteins, fats, carbohydrates), proteins are more susceptible to the influence of the temperature factor. This is due to the high risk of heat as a damaging factor for proteins and the evolutionary protective strategy of the animal organism, which is associated with an increase in the production of heat shock proteins. The increased expression of heat shock proteins leads to a decrease in the expression and synthesis of other proteins, which we observed in the decrease in the level of protein, including milk caseins, in the summer. At the same time, maintaining a stable level of lactose synthesis under conditions of heat stress remains constant. This indicates that the importance of this component of milk is comparable to the importance of heat shock proteins. Our study provides some information about goat milk in different seasons and shows the influence of the temperature factor on the biochemistry of milk and its antioxidant composition. When processing goat milk, it is worth considering this and planning the production of products with desired functional properties while taking into account seasonal patterns of changes in the composition of goat milk.

## Figures and Tables

**Figure 1 animals-13-01706-f001:**
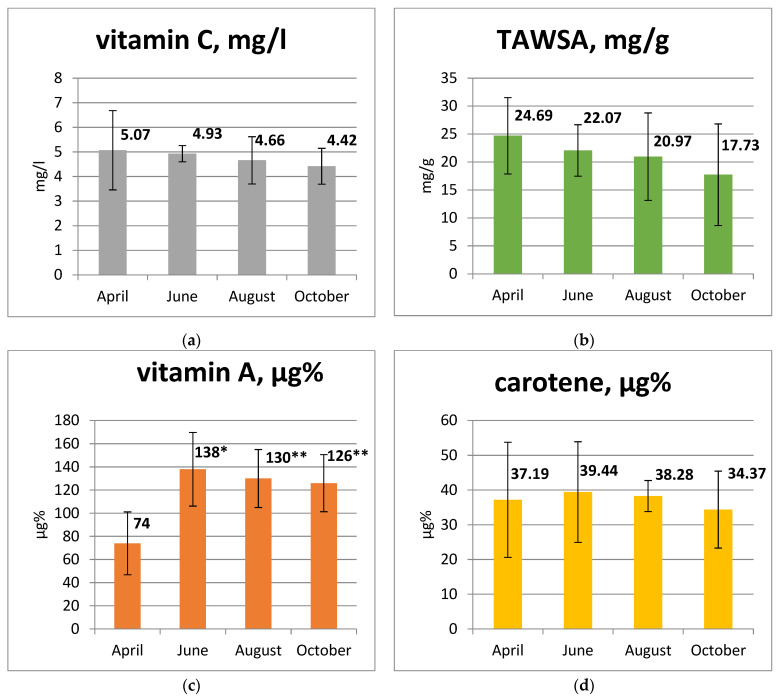
(**a**) Content of vitamin C in goat milk; (**b**) TAWSA means total amount of water-soluble antioxidants; (**c**) the content of vitamin A in goat milk; (**d**) carotene content in goat milk. Note: * *p* ˂ 0.01; ** *p* ˂ 0.001.

**Figure 2 animals-13-01706-f002:**
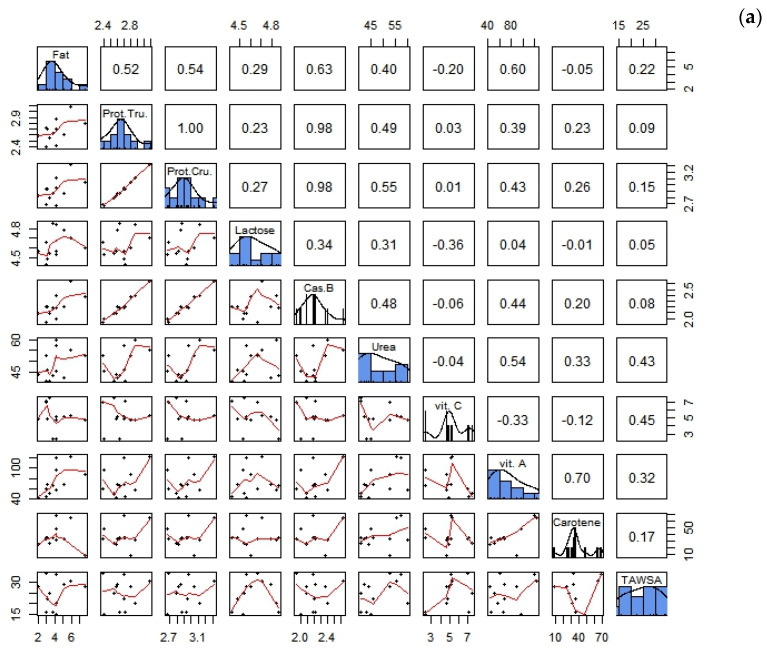
The most important correlations between all the data studied: (**a**) in April; (**b**) in June; (**c**) in August; (**d**) in October.

**Table 1 animals-13-01706-t001:** Biochemical parameters of goat milk depending on the season.

Parameters	April	June	August	October
Mass fraction of fat MFF (%)	4.14 ± 0.43	3.80 ± 0.23	3.94 ± 0.13	4.65 ± 0.24
Mass fraction of true protein MFTP-1 (%)	2.67 ± 0.06	3.06 ± 0.07 (*p* ≥ 0.01) *	3.17 ± 0.12 (*p* ≥ 0.05) *	4.37 ± 0.15
Mass fraction of crude protein MFCP-2 (%)	2.92 ± 0.05 (*p* ≥ 0.01) *	3.28 ± 0.06 (*p* ≥ 0.001) *	3.37 ± 0.11 (*p* ≥ 0.01) *	4.44 ± 0.14
Lactose (%)	4.63 ± 0.04	4.44 ± 0.05	4.35 ± 0.07	4.41 ± 0.07
Dry skimmed milk matter DSMM (%)	8.17 ± 0.10 (*p* ≥ 0.05) *	8.42 ± 0.10 (*p* ≥ 0.001) *	8.39 ± 0.16 (*p* ≥ 0.001) *	9.75 ± 0.14
Total dry matter TDM (%)	12.33 ± 0.47	12.24 ± 0.26	12.34 ± 0.21 (*p* ≥ 0.001) *	14.24 ± 0.31
Caseins (%)	2.21 ± 0.06 (*p* ≥ 0.01) *	2.51 ± 0.06 (*p* ≥ 0.001) *	2.56 ± 0.09 (*p* ≥ 0.001) *	3.55 ± 0.12 (*p* ≥ 0.001) *
Aceton (mM/L)	0.15 ± 0.03 (*p* ≥ 0.001) *	0.05 ± 0.03	0.08 ± 0.02	0 ^&^
Betahydroxybutyrate BHB (mM/L)	0.06 ± 0.01 (*p* ≥ 0.01) *	0.01 ± 0.02	0.03 ± 0.01 (*p* ≥ 0.001) *	0 ^&^
Urea (mg/100 mL)	48.51 ± 0.94 (*p* ≥ 0.001) *	51.08 ± 3.31	43.01 ± 2.01	56.40 ± 2.49
Freezing point (°C)	−0.547.25 ± 0.54 (*p* ≥ 0.001) *	−0.554 ± 0.22	−0.562 ± 0.19	−0.567 ± 0.24
pH (a.u. ^#^)	6.60 ± 0.02 (*p* ≥ 0.001) *	6.57 ± 0.02	6.53 ± 0.03 (*p* ≥ 0.001) *	6.61 ± 0.03

Notes: (*p* ≥ 0.05–0.001) * the reliability of data differences (each month compared to October); a.u. ^#^—arbitrary units; ^&^ within the experimental error.

## Data Availability

Data supporting reported results can be found at https://www.vij.ru/goszadanie-i-proekty/proekty/proekty-rnf (accessed on 12 May 2023).

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
