# Peer review of "Seasonal Changes in the Antioxidant Activity and Biochemical Parameters of Goat Milk"

_animals, 2023, doi:10.3390/ani13101706_

Round 1
Reviewer 1 Report
1. The paper suffers from a poor English structure throughout. The manuscript requires thorough proofreading by a native person whose first language is English.
2. The novelty of the study needs to be highlighted compared to other similar studies.
3. In material and method: Heat stress is mentioned in the article, but climate data was not included in the article. This data should be added to the article and associated with the milk data.
-The trial plan is unclear. The Material Method part needs to be rewritten.
The animal material used in the article is not fully unexplained. Therefore, many questions come to mind. Some of these questions are;
-Were all the animals used in the experiment of the same parity?
-How old are the animals?
-The differences in milk content are due to different lactation periods or seasonal differences?
3. Discussion is weak. The discussion needs enhancement with real explanations, not only agreements and disagreements. Authors should improve it by the demonstration of biochemical/physiological causes of obtained results. Instead of just justifying results, results should be interpreted and explained to appropriately elaborate inferences. The discussion seems to be poor, didn't give good explanations of the results obtained. I think that it must be really improved. Where possible please discuss potential mechanisms behind your observations. You should also expand the links with prior publications in the area but try to be careful to not over-reach. For the latter, you should highlight potential areas of future study.
4. The conclusion part was similar to the discussion. However, this section is where you should present your conclusions from this article. This part needs to be rewritten.
The paper suffers from a poor English structure throughout. The manuscript requires thorough proofreading by a native person whose first language is English.
Author Response
Dear Reviewer,
The authors agree with all reviewer's comments and made the numerous corrections according to reviewer's comments. In particular,
- We agree that “the paper suffers from a poor English structure throughout”. We have got the professional help on this aspect from our colleagues with strong knowledge in English. If you find that this manuscript (ID: animals-2359978 by Oksana A. Voronina et al.) after the revision will need more editing, we will be ready to use the professional Editing service (https://www.editage.com).
- The novelty of the study (in comparison with other similar studies) is associated with an integrated approach to the study of the biochemical and antioxidant composition of goat milk depending on the season for various breeds of goats, including local Russian breeds. This work is especially important for climatic regions with similar cycles of temperature change. The relevance of this work is associated with changes in climatic conditions and increased interest in the adaptation of productive animals to these changes.
We studied in details a number of similar works [refs. 4,6,17-25], including the works that describe in more depth the mechanism of the effect of temperature increase on the quality of goat milk and changes in its biochemical composition [refs. 23-25]. There are the optimal temperature zones for goats and indicated what circumstances associated with an increase in temperature activate their adaptive systems. In this regard, we added two paragraphs to the introduction on pages 1 and 2, where we tried to highlight the advantages of our work in comparison with the others. In addition, we focused on describing the relationship between the antioxidant properties of goat milk and the biochemical composition in its complex colloidal system. For example, the total amount of water-soluble antioxidants in goat milk was studied by us for the first time.
- In section 2. “Materials and methods” we added subsection 2.1. “Climatic conditions”. In this subsection 2.1. the average monthly temperatures are given for the region where sampling have been carried out for the whole last year (see page 4). We clarified the test plan, indicated the breed composition of the groups, the age and timing of lactation. Additionally, we indicated that before making a conclusion about the influence of the season, all other conditions: the influence of breed, age, and timing of lactation were excluded. The mentioned above is described on page 4 (in paragraph 2.2.): sampling of goat milk and statistical processing of the results.
- Substantial changes have been made to section 4 of the “Discussion”. We have described all this in the part 4.1, including a comparison of our results with the results of some previous works. In addition, we have been prepared a separate subsection 4.2. “Physiological and biochemical mechanisms of goat adaptation to heat stress” (pages 10-12). Here we have reinforced our data by explaining the molecular mechanisms of adaptation to elevated temperatures. In addition, we have made more attention to the physiological and hormonal adaptation in the part “Introduction”. In the included refs. [6,7,8], we have tried to reveal some molecular mechanisms of metabolic changes at the cellular level. To do this, we relied on a number of works revealing the genetic component of the process of adaptation to heat stress [refs. 35-42] and expanded the references to publications by adding a total of the 14 literature sources. In this section, we have briefly touched on the physiological and biochemical reasons for the results obtained, and outlined areas for further research. We have tried to interpret and to explain the results obtained, as well as – to draw the appropriate conclusions.
- The final part of the “Conclusions” has been completely rewritten. We have tried to summarize and provide full-fledged conclusions on the work done in this publication (page 12).
In conclusion, we are grateful to the Reviewer for the valuable comments, which allowed us to reveal the details of the experiment and to discuss the whole work in a better way.

Reviewer 2 Report
I reviewed the manuscript (animals-2359978) entitled: Seasonal changes in antioxidant activity and biochemical parameters of goat milk.
Strengths: The study describes an interesting subject, investigating the antioxidant activity of goat milk in relation to the changes in the main biochemical parameters depending on the season.
The information presented were new and the conclusions supported by the data.
Some specific points.
Line 95. Please clarify abbreviations SCWA and SCVA.
Lines 295-300. The paragraph should move to the introduction section.
Author Response
Dear Reviewer,
The authors agree with all reviewer's comments and made the numerous corrections according to reviewer's comments. In particular,
- The authors clarified abbreviations “the total amount of water soluble antioxidants” as TAWSA
(instead of SCWA and SCVA) all over the text.
- The authors rewrote all text (from the “Introduction” to the “Conclusions”).
We are grateful to the Reviewer for the valuable comments, which allowed us to reveal the details of the experiment and to discuss the whole work in a better way.
Sincerely Yours,
Sergei Zaitsev
Oksana Voronina

Round 2
Reviewer 1 Report
When I examined the article, I saw that the deficiencies I mentioned earlier were corrected. It can be published in your journal as it is.